# Impact of Treating Age-Related Macular Degeneration before Visual Function Is Impaired

**DOI:** 10.3390/jcm11195726

**Published:** 2022-09-27

**Authors:** Risa Aichi, Norihiro Nagai, Kishiko Ohkoshi, Yoko Ozawa

**Affiliations:** 1Department of Ophthalmology, St. Luke’s International Hospital, 9-1 Akashi-cho, Chuo-ku, Tokyo 104-8560, Japan; 2St. Luke’s International University, 9-1 Akashi-cho, Chuo-ku, Tokyo 104-8560, Japan; 3Department of Ophthalmology, Keio University School of Medicine, 35 Shinanomachi, Shinjuku-ku, Tokyo 160-8582, Japan

**Keywords:** age-related macular degeneration, anti-vascular endothelial growth factor treatment, visual outcome, Kaplan–Meier survival analysis

## Abstract

Visual outcomes of age-related macular degeneration (AMD) have substantially improved via anti-vascular endothelial growth factor (anti-VEGF) therapy. However, the treatment effects vary among individuals. Medical charts of 104 eyes (104 patients) with AMD, treated with anti-VEGF drugs and followed up for 12–36 months, were retrospectively analyzed. Logistic regression analyses adjusted for age showed that eyes with an initial best-corrected visual acuity (BCVA) < 0.3 in the logarithm of the minimum angle of resolution (logMAR) were a positive predictor (odds ratio = 3.172; 95% confidence interval [CI] = 1.029–9.783; *p* = 0.045), and the presence of initial fibrovascular pigment epithelial detachment (PED) was a negative predictor (0.222; 0.078–0.637; *p* = 0.005) of maintained or improved BCVA at the final visit. Kaplan–Meier survival analysis showed that eyes with an initial BCVA < 0.3 (Cox hazard ratio = 2.947; 95% CI = 1.047–8.289; *p* = 0.041) had a better survival rate after adjusting for age when failure was defined as a BCVA reduction ≥ 0.2 of logMAR. Eyes with an initial BCVA < 0.3 belonged to younger patients; more frequently had subretinal fluid as an exudative change; and less frequently had intraretinal fluid, submacular hemorrhage, and fibrovascular PED. Initiating anti-VEGF treatment before BCVA declines and advanced lesions develop would afford better visual outcomes for AMD eyes in the real-world clinic, although further analyses are required.

## 1. Introduction

Age-related macular degeneration (AMD) is a vision-threatening disease, affecting 8.4 million people worldwide [1]. Although the development of anti-vascular endothelial growth factor (anti-VEGF) treatment has substantially improved overall visual outcomes [2,3,4,5], these vary across patients in real-world practice. Evaluating the treatment outcomes and factors associated with a good prognosis may help clinicians improve their treatment plans.

In most phase III clinical trials, additional anti-VEGF treatments have been performed proactively and frequently following predefined protocols. In the VIEW 1 and 2 trials, 95 to 96% of patients achieved maintained or improved best-corrected visual acuity (BCVA) 2 years after the initial injection of either ranibizumab or aflibercept with a proactive treatment protocol, such as every 1 or 2 months [5]. However, additional treatments after the initial injection were granted to the individual clinicians and patients in the real-world clinic. Owing to economical and/or other reasons, treatment regimens such as pro re nata [6,7] and treat-and-extend [8] are now applied to clinical practice to reduce the treatment and injection numbers of anti-VEGF drugs. Although both treatment regimens require the judgement of disease activity by individual clinicians, and the determinations of initiation and additional treatment are not unified among clinicians, recent real-world studies show that the treatment is mostly successful [9,10,11,12,13]. However, these analyses were performed using the data of patients who were fully followed up during the course, and many of those who dropped out were excluded; this may not necessarily represent the real-world practice.

Here, we retrospectively reviewed the visual outcomes of eyes with AMD treated with anti-VEGF drugs in a real-world setting. The current study involved all patients who received anti-VEGF treatment as the first therapy for AMD, and were followed up for at least 12 months and up to 36 months, including those who dropped out during the study period; patients who dropped out were included in Kaplan–Meier survival analysis, which may more precisely reflect real-world conditions of the daily clinical practice.

Our results may help elucidate points requiring attention when examining eyes with AMD and the timing for offering treatment to this patient population in real-world settings. They may also emphasize the importance of advising individuals to visit the eye clinic when they observe even minor symptoms and clinicians to detect AMD lesions at the early stage.

## 2. Materials and Methods

The procedures of this retrospective study adhered to the tenets of the Declaration of Helsinki, and the study protocol was approved by the St. Luke’s International University Ethics Committee (approval number: 20-R048). Informed consent was obtained from all patients involved in the study.

### 2.1. Patients

Medical charts of patients diagnosed with neovascular AMD between January 2017 and August 2020 in the Department of Ophthalmology, St. Luke’s International Hospital (Tokyo, Japan), treated with anti-VEGF drugs as the first therapy for AMD, and followed up for longer than 12 months, were retrospectively analyzed. Those who were treated by other therapies than anti-VEGF drugs, such as SF6 injection for massive subretinal hemorrhage, or who had no fluorescein angiography for AMD diagnosis, were excluded. The data were collected from up to 36 months of follow-up. There were no patients with both eyes treated during this period.

### 2.2. Eye Examinations

All patients underwent complete ophthalmologic examinations, including BCVA measurement with a refraction test, slit-lamp examinations, and indirect ophthalmoscopy after pupil dilation with 0.5% tropicamide. BCVA measured using Landolt C charts was converted into the logarithm of the minimum angle of resolution (logMAR) by calculating log (10) of the decimal score obtained by the Landolt C chart for the purposes of statistical analyses. Fluorescein and indocyanine green angiographies were recorded using a Heidelberg Spectralis HRA instrument (Heidelberg Engineering GmbH, Dossenheim, Germany). Optical coherence tomography (OCT) images were obtained at every follow-up visit using a Cirrus HD-OCT system (Zeiss, Oberkochen, Germany) or a Heidelberg Spectralis OCT system (Heidelberg Engineering GmbH). The improvement or loss of BCVA compared with the initial BCVA was defined when the BCVA changed by ≥ 0.2 in logMAR.

### 2.3. Treatments

Patients were treated with anti-VEGF drugs: both or either intravitreal ranibizumab 0.5 mg and/or aflibercept 2 mg injections. Additional injections after the initial injection, and changes in the anti-VEGF drugs were determined by each clinician. One patient was treated with photodynamic therapy, and one received laser treatment after the initial anti-VEGF therapy according to the clinicians’ determinations.

### 2.4. Definition of Treatment Failure (Death)

Treatment failure was defined as a BCVA reduction ≥ 0.2 of logMAR compared with the initial BCVA at each time point. Failure was also considered when patients received treatment other than anti-VEGF as described above. Survival or failure was determined every 6 months after the initial anti-VEGF treatment. 

### 2.5. Statistical Analyses

Data are expressed as the mean ± standard deviation. Analyses with the Wilcoxon rank sum test, logistic regression analyses, Kaplan–Meier survival analysis, Cox proportional hazards model, Mann–Whitney U test, and chi-square test were performed using SPSS (version 25.0; SPSS Japan, Tokyo, Japan). Non-parametric analyses were performed. A *p* value < 0.05 was considered statistically significant.

## 3. Results

### 3.1. Patients’ Demographics

For the 104 eyes of 104 patients with AMD, the mean age was 73.8 ± 1.1 (range, 50 to 93; median, 77) years, and 69 (66.3%) of the patients were men (Table 1). Eighty-five eyes (81.7%) were categorized as having polypoidal choroidal vasculopathy, thirteen eyes (12.5%) had typical AMD, and seven eyes (6.7%) had retinal angiomatous proliferation. The mean BCVA was 0.444 ± 0.052 (range, −0.176–2.3; median, 0.301) in logMAR, and the mean central retinal thickness measured in the OCT images was 356 ± 16 μm (range, 102–911 μm; median, 309). The mean BCVA at the final visit was 0.360 ± 0.049 (range, −0.176–2.0; median, 0.155) in logMAR, and there was significant improvement according to the Wilcoxon rank sum test (*p* = 0.012). Maintained or improved BCVA at the final visit ≥ 0.2 in logMAR compared with initial BCVA was observed in 93 eyes (89.4%). The mean injection number was 7.5 ± 0.5 (range 1 to 22).

### 3.2. Predictive Factors for Good Visual Outcome

Predictive factors for good visual outcome, i.e., a maintained or improved BCVA at the final visit ≥ 0.2 in logMAR compared with the initial BCVA, were assessed with logistic regression analyses adjusted for age. A BCVA < 0.3 in logMAR at baseline was a positive factor (odds ratio (OR), 3.172; 95% confidence interval [CI], 1.029–9.783; *p* = 0.045), and the presence of fibrovascular pigment epithelial detachment (PED) was a negative factor (OR, 0.806; 95% CI, 0.078–0.637; *p* = 0.085) for maintaining or achieving better BCVA at the last visit (Table 2). Scatter plots showing the relationships between the initial and final BCVA in individuals, visualized that the BCVA at baseline was correlated with the final BCVA after anti-VEGF treatment (Figure 1).

### 3.3. Kaplan–Meier Survival Analysis after Anti-Vascular Endothelial Growth Factor Treatment

The eyes were divided into two groups according to whether they had a BCVA < 0.3 in logMAR, i.e., the median BCVA in the current study. The Kaplan–Meier survival analysis showed that of the 51 eyes that exhibited an initial BCVA < 0.3 and 53 eyes that exhibited an initial BCVA ≥ 0.3 in logMAR, 89.8% (mean survival time, 33.4 months; 95% CI, 31.3–35.6) and 58.0% (mean survival time, 28.4 months; 95% CI, 25.0–31.8) had, respectively, survived at 36 months when failure was defined as a BCVA reduction ≥ 0.2 from the initial BCVA at each time point (Figure 2). The eyes with an initial BCVA < 0.3 in logMAR had a significantly better survival rate by the logrank test (*p* = 0.014). The survival rate at 6 months of the group with an initial BCVA < 0.3 in logMAR was 98.0% and that of the group with an initial BCVA ≥ 0.3 in logMAR was 80.4%; note that approximately half of failure cases had been observed by 6 months in the worse BCVA group.

The reason for failure in two eyes (40%) was due to additional treatment other than anti-VEGF and in three eyes (60%), there was decreased BCVA ≥ 0.2; one eye exhibited fibrovascular PED, one exhibited hemorrhage, and one exhibited serous PED in the group with an initial BCVA < 0.3. In the group with an initial BCVA ≥ 0.3, two eyes (12.5%) failed owing to additional treatment other than anti-VEGF and fourteen eyes (87.5%) failed owing to decreased BCVA; eight eyes exhibited fibrovascular PED, five exhibited subretinal fluid (SRF), and one exhibited intraretinal fluid (IRF) (Figure 3).

The number of eyes analyzed at each time point was shown below the graph of the Kaplan–Meier survival analysis (Figure 2). Patients who dropped out by 36 months more frequently involved those whose BCVA > 0.5 at baseline (*p* = 0.039) and whose BCVA > 0.5 at the last visit (*p* = 0.025) (Table 3). There were no differences in age and sex between the patients who dropped out or not.

### 3.4. Conditions for Treatment Success

The baseline characteristics of the eyes related to survival or failure were analyzed using Cox proportional hazard models adjusted for age (Table 4). an initial BCVA < 0.3 in logMAR (hazard ratio, 2.947; 95% CI, 1.047–8.289; *p* = 0.041) was related to better survival probability. 

The baseline characteristics of the eyes with or without an initial BCVA < 0.3 in logMAR were further compared. Eyes with an initial BCVA < 0.3 in logMAR belonged to younger patients (*p* = 0.008) and less frequently had IRF (*p* < 0.001), SRF (*p* = 0.003), submacular hemorrhage > 1 disc in diameter (*p* = 0.030), and fibrovascular PED (*p* = 0.016), all at baseline (Table 5). Among the eyes with SRF, only 1 of 49 (2.0%) with an initial BCVA < 0.3 exhibited IRF, while as many as 11 of 40 eyes (27.5%) with an initial BCVA ≥ 0.3 exhibited IRF, and the difference was significant (*p* = 0.0005).

## 4. Discussion

We reported real-world clinical data of visual outcomes after anti-VEGF treatment for AMD. Among the 104 eyes of 104 patients with neovascular AMD who were followed up for at least 12 months and up to 36 months, the median initial BCVA was 0.3. According to logistic regression analyses adjusted for age, having an initial BCVA < 0.3 in logMAR was a positive predictor, and the initial presence of fibrovascular PED was a negative predictor of maintained or improved BCVA at the final visit after anti-VEGF treatment. Kaplan–Meier survival analysis showed that the eyes with an initial BCVA < 0.3 in logMAR had a better survival rate than the others when failure was defined as a BCVA reduction ≥ 0.2 from the initial BCVA at each time point up to 36 months. Eyes with an initial BCVA < 0.3 in logMAR belonged to younger patients; more frequently exhibited SRF; and less frequently exhibited IRF, submacular hemorrhage, and fibrovascular PED at baseline.

Given that the inclusion criterion of prospective clinical trials, such as VIEW 1 and 2, was 73 to 25 (20/40 to 20/320) letters in BCVA, measured using an ETDRS chart, the median BCVA of 0.3 in logMAR, corresponding to 0.5 in the Landolt C chart and 20/40 in the Snellen chart, was a relatively good BCVA for the eyes before receiving anti-VEGF treatment; treatment was initiated rather more actively than in the clinical trials in the current real-world study from 2017 to 2020. Overall, the mean BCVA improved in the current study, suggesting that the treatment was largely successful. Consistently, significant BCVA improvement was achieved after anti-VEGF treatments in our previous retrospective study of patients with PCV treated by a pro re nata regimen at 2 years [9], and another study of patients with neovascular AMD treated by a treat-and-extend regimen at 3 years [14]. However, BCVA improvement was not significant in a multicenter retrospective study at 3 years, while it was significant at 1 year [15], and a less than 1 ETDRS letter improvement was obtained in another multicenter study at 6 years [16]. Because the Kaplan–Meier analysis in the current study showed that the patients whose BCVA decreased > 0.2 in logMAR increased over time, a longer follow-up study would be of value in the future.

An initial BCVA < 0.3 in logMAR was a predictive factor for achieving maintained or improved BCVA with an OR of 3.152. This result was consistent with that reported by a previous prospective study by Minami et al. where the visual prognosis of patients with AMD whose initial BCVA was < 0.22 in logMAR, corresponding to a BCVA > 74 letters in the ETDRS chart, was analyzed, showing overall improvement in BCVA with anti-VEGF monotherapy [10]. There was no concern of a ceiling effect in BCVA after treatment while the patients had a relatively good initial BCVA. In fact, the best BCVA before anti-VEGF treatment was −0.176 in logMAR (20/13 in the Snellen chart) in the current study. The previous report by Minami et al. was a single-arm study and did not compare patients who had a worse initial BCVA [10]. However, the current study revealed that the treatment outcomes of eyes with an initial BCVA < 0.3 in logMAR were significantly better than those of eyes with a worse initial BCVA, putting forth the recommendation of treating AMD while the BCVA remains good.

Interestingly, the eyes with an initial BCVA < 0.3 in logMAR less frequently experienced a BCVA decline of ≥ 0.2 in logMAR according to the Kaplan–Meier survival analysis. Patients with eyes with an initial BCVA < 0.3 in logMAR were younger, suggesting that these patients may have had a greater biological availability for tissue function restoration, given that older individuals are affected by age-related neuronal vulnerability [17]. All eyes had exudative changes such as IRF and/or SRF at baseline, which comprised the reason for initiating anti-VEGF treatment. However, the eyes with an initial BCVA < 0.3 in logMAR less frequently exhibited IRF, exhibiting only SRF. This is consistent with the findings of a previous report where eyes with IRF at baseline had worse visual outcomes [18]. IRF may have been developed from macular neovascularization (MNV) in the choroid thorough the retinal pigment epithelium (RPE) and retinal tissue, suggesting that the influence of the MNV could have spread widely in the retinal layers if they already had IRF, potentially developed by intensive exudation with or without a longer period of changes; therefore, retinal neuronal damage became greater. It has also been reported that not all intraretinal cystoid spaces recorded on OCT images, considered IRFs, are caused by active MNV-associated exudation, and the intraretinal cystoid space may be associated with neurosensory degeneration, also termed a degenerative intraretinal cystoid cavity [18]. We previously reported that in diabetic macular edema, eyes with a greater area of initial IRF did not achieve improvement of visual function but had retinal thinning after IRF resolution via anti-VEGF treatment, most likely because of reduction in the retinal neural components and retinal neurodegeneration [19] related to persistent IRF. Additionally, the eyes with an initial BCVA < 0.3 in logMAR less frequently had initial submacular hemorrhage. Hemorrhage was closely associated with a poor visual prognosis because of irreversible damage to the photoreceptors [20,21]. Massive submacular hemorrhage is treated with an intravitreal tissue plasminogen activator and gas injections for the purpose of replacing the hemorrhage to avoid further damage to the foveal photoreceptors [22,23]. However, the same procedures are not applied to minor hemorrhage, which may easily cause photoreceptor damage before treatment.

The presence of fibrovascular PED at baseline was a risk factor for BCVA decline after anti-VEGF treatment. Fibrovascular PED was also a major reason for visual loss, and eyes with worse BCVA (≥ 0.3 in logMAR) more frequently had fibrovascular PED. This is consistent with a report that non-responding eyes, without improvement in BCVA and/or exudative changes after anti-VEGF treatment, often have fibrovascular PED [12,13]. Fibrovascular PED reportedly involves a multilayered lamellar scar between the choroid and RPE, suggesting that the transfer of nutrients and oxygen from the choroid to the RPE and photoreceptors may be disrupted, resulting in RPE and photoreceptor dysfunctions. Although the RPE has a blood–retinal barrier function and functions as a fluid transfer system from the retinal to the choroidal side, RPE disorders may have easily increased the fluid above the RPE and decreased fluid absorption. Moreover, photoreceptor function may have already been damaged owing to the presence of fibrovascular PED-related malnutrition as described above. In fact, eyes with an initial BCVA ≥ 0.3 in logMAR frequently had fibrovascular PED. Further, fibrovascular PED may disrupt anti-VEGF drug transfer to the choroidal lesion. Therefore, fibrovascular PED may be related to the insufficient effect of anti-VEGF therapy, as well as neurodegeneration already present in the retina, both of which may cause a worse visual outcome. Meanwhile in the presence of fibrovascular PED, it has been reported that some eyes, which may not respond to a significant anatomical reduction in PED height, may still sustain visual gains or achieve visual stability with anti-VEGF therapy [24]. Appropriate informed consent would be required from patients with fibrovascular PED at baseline.

IRF was not a significant predictive factor for the risk of treatment failure, although the presence of initial IRF was related to worse BCVA at baseline; this may be explained by the small number of eyes with initial IRF in the current study.

We analyzed the clinical data using Kaplan–Meier survival analysis. This enabled us to also include the eyes of patients who dropped out during the follow-up, potentially reflecting real-world clinical practice conditions. Patients who dropped out during the follow-up involved both those who had a good course and were satisfied, with or without reverse referral to local clinics, and those who had abandoned treatment owing to the absence of improvement or an unsatisfactory outcome. Clinicians cannot predict whether any patient will or will not drop out during the follow-up; however, according to the current study, eyes with an initial BCVA < 0.3 in logMAR could experience sustained or improved BCVA with treatment initiation. In addition, almost half of failure cases were observed within the first 6 months in eyes with an initial worse BCVA. This information could be used to encourage patients to initiate anti-VEGF treatment while their BCVA remains relatively good.

The limitations of the current study included a relatively small number of patients, retrospective analysis design, involving eyes treated with either or both ranibizumab and aflibercept, and that additional treatment was determined by each clinician. However, this would reflect real-world clinical practice conditions involving variations in clinicians’ determinations over treatment, and the data would be informative to general clinicians. We adjusted only age in the logistic regression analyses considering the sample size. Because some of the patients used both drugs, we did not analyze the results according to the drug selection. All reasons for dropping out were not documented; however, the Kaplan–Meier survival analysis covered this bias.

## 5. Conclusions

In conclusion, eyes with a relatively good initial BCVA, such as < 0.3 in logMAR, and without fibrovascular PED would achieve sustained or improved BCVA after anti-VEGF treatment within 3 years of follow-up in the real-world setting. Early treatment would be recommended for exudative AMD, although further studies are required.

## Figures and Tables

**Figure 1 jcm-11-05726-f001:**
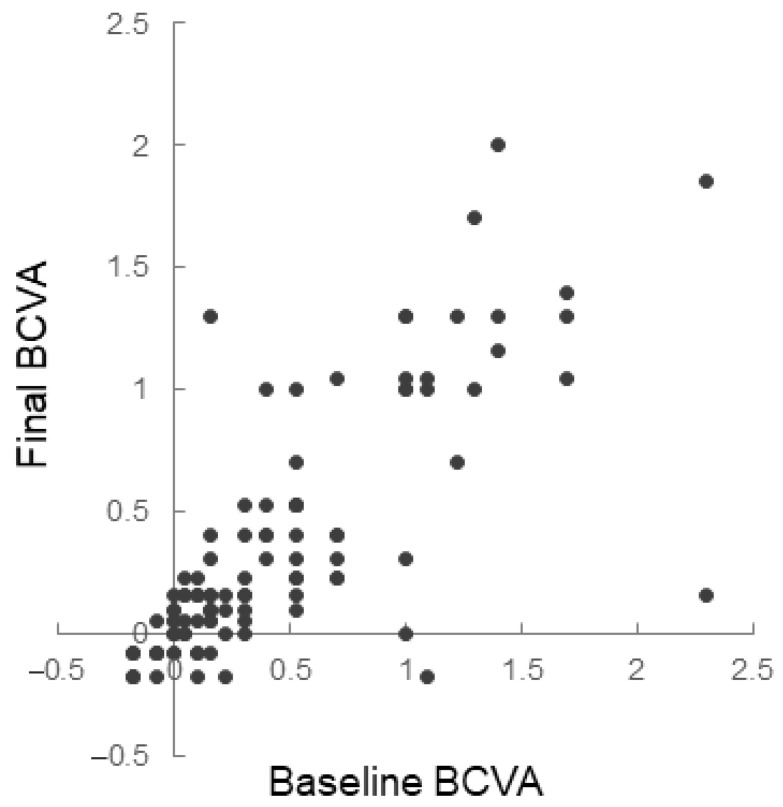
Scatter plots of baseline and final best-corrected visual acuity (BCVA) of the individual eyes. BCVA is expressed after conversion to the logarithm of the minimum angle of resolution.

**Figure 2 jcm-11-05726-f002:**
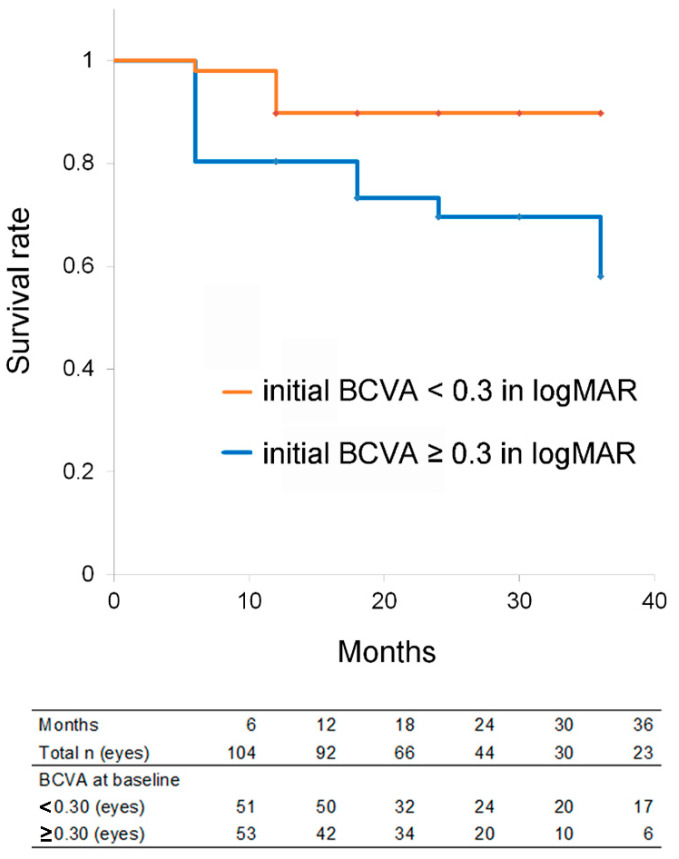
The Kaplan–Meier survival analysis after anti-vascular endothelial growth factor treatment in age-related macular degeneration. Failure was defined as a best-corrected visual acuity (BCVA) reduction ≥ 0.2 of the logarithm of the minimum angle of resolution (logMAR) from the initial BCVA. The eyes with an initial BCVA better than 0.3 in logMAR (shown in orange) had a better survival rate than those with an initial BCVA equal or worse than 0.3 (shown in blue). *p* = 0.014 by logrank test. The number of eyes at each time point is shown.

**Figure 3 jcm-11-05726-f003:**
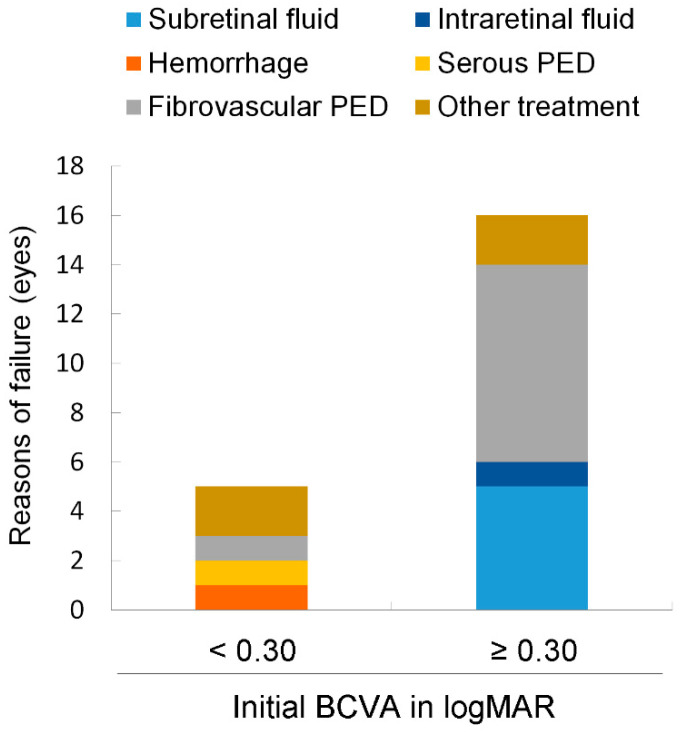
The conditions at the time of failure. Failure of the eyes was determined by a decline of best-correlated visual acuity (BCVA) ≥ 0.2 from the initial BCVA or by the application of treatments other than anti-vascular endothelial growth factor. The numbers of eyes exhibiting fundus findings most likely related to BCVA loss are shown. Two eyes failed owing to other treatment application in both groups, i.e., with or without an initial BCVA < 0.3 in the logarithm of the minimum angle of resolution (logMAR). PED, pigmental epithelium detachment.

**Table 1 jcm-11-05726-t001:** Baseline characteristics.

Eyes	104
Age (years)	73.8 ± 1.1 (range, 50–93)
Sex (men; eyes (%))	69 (66.3%)
Subtypes of age-related macular degeneration (AMD)	
Polypoidal choroidal vasculopathy (PCV)	84 (80.8%)
Typical AMD	13 (12.5%)
Retinal angiomatous proliferation (RAP)	7 (6.7%)
Best-corrected visual acuity (logMAR)	0.444 ± 0.053 (−0.176–2.3)
Central Retinal Thickness (μm)	356 ± 16 (102–911)

Data are shown as the mean ± standard deviation.

**Table 2 jcm-11-05726-t002:** Predictive factors for maintained or improved best-corrected visual acuity at the last visit, after adjusting for age.

	Odds Ratio	95% Confidence Interval	*p*
Sex (Male)	0.791	0.274–2.279	0.664
Best-corrected visual acuity <0.3 in logMAR	3.172	1.029–9.783	0.045 *
Subtypes of age-related macular degeneration (AMD)			
Polypoidal choroidal vasculopathy	1.729	0.547–5.643	0.351
Typical AMD	0.723	0.169–3.106	0.663
Retinal angiomatous proliferation	0.500	0.098–2.554	0.405
Central retinal thickness ≥ 300 μm	1.231	0.457–3.313	0.681
Intraretinal fluid (IRF)	0.806	0.261–2.490	0.708
Subretinal fluid (SRF)	0.876	0.211–3.644	0.856
All types of hemorrhages	1.499	0.559–4.476	0.388
Sub-macular hemorrhage (> 1 disc diameter)	0.409	0.101–1.651	0.209
Serous pigment epithelial detachment (PED)	1.288	0.429–3.863	0.652
Hemorrhagic PED	0.871	0.246–3.087	0.831
Fibrovascular PED	0.222	0.078–0.637	0.005 **
Subretinal hyper-reflective material (SHRM)	0.871	0.246–3.087	0.831

Logistic regression analyses adjusted for age. * *p* < 0.05, ** *p* < 0.01.

**Table 3 jcm-11-05726-t003:** Characteristics of the patients who dropped out before month 36.

	Dropped Out	Continued	*p*
n (Eyes (%))	23 (22.1)	81 (77.9)	
Age (mean ± standard deviation (range))	75.5 ± 2.5 (56–91)	73.4 ± 1.2 (50–93)	0.361
Sex (male %)	14 (60.8)	55 (67.9)	0.866
BCVA at baseline >0.5 (Eyes (%))	13 (56.5)	27 (33.3)	0.039 *
BCVA at last visit >0.5 (Eyes (%))	11 (47.8)	18 (22.2)	0.025 *

Mann–Whitney U tests and Chi-square tests. BCVA, best-corrected visual acuity. * *p* < 0.05.

**Table 4 jcm-11-05726-t004:** Cox hazard ratios of factors associated with post-therapeutic maintained or improved best-corrected visual acuity.

	Hazard Ratio	95% Confidence Interval	*p*
Sex (Male)	0.965	0.389–2.393	0.939
Best-corrected visual acuity < 0.3 in logMAR	2.947	1.047–8.289	0.041 *
Polypoidal choroidal vasculopathy (PCV)	1.930	0.747–4.988	0.175
Central retinal thickness ≥ 300 μm	1.325	0.563–3.123	0.519
Intraretinal fluid (IRF)	0.841	0.321–2.204	0.724
Sub-macular hemorrhage (> 1 disc diameter)	0.480	0.161–1.432	0.188

Cox proportional hazard models adjusted for age. * *p* < 0.05.

**Table 5 jcm-11-05726-t005:** Characteristics of patients with better best-corrected visual acuity (BCVA) (< 0.3 in logMAR) at baseline.

	BCVA at Baseline (logMAR)		*p*
	<0.30	≥0.30	
**Eyes**	51	53	
Age (years)	70.9 ± 1.5 (50–93)	76.6 ± 1.5 (51–92)	0.008 **
Sex (men; eyes (%))	37 (72.5)	32 (60.4)	0.134
Subtypes of age-related macular degeneration (AMD)			
Polypoidal choroidal vasculopathy	43 (84.3)	42 (77.4)	0.135
Typical AMD	4 (13.7)	6 (11.3)	
Retinal angiomatous proliferation	1 (2.0)	6 (11.3)	
Central Retinal Thickness (μm)	342 ± 23 (102–883)	370 ± 24 (153–911)	0.309
Subretinal fluid (SRF) (eyes (%))	49 (96.1)	40 (75.5)	0.003 **
Intraretinal fluid (IRF) (eyes (%))	2 (3.9)	21 (39.6)	<0.001 **
All types of hemorrhages (eyes (%))	16 (31.4)	26 (49.1)	0.050
Submacular hemorrhage (> 1 disc diameter) (eyes (%))	2 (3.9)	9 (17.0)	0.030 *
Hemorrhagic PED (eyes (%))	8 (15.7)	12 (22.6)	0.258
Serous PED (eyes (%))	18 (35.3)	12 (22.6)	0.114
Fibrovascular PED (eyes (%))	8 (15.7)	19 (35.8)	0.016 *
Subretinal hyper-reflective material (SHRM) (eyes (%))	17 (33.3)	19 (35.8)	0.475

Data are shown as the mean ± standard deviation. Mann–Whitney U tests and Chi-square tests. * *p* < 0.05, ** *p* < 0.01.

## Data Availability

Data are available upon request to the corresponding author with appropriate reasoning.

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
