# Peer review of "Impact of Treating Age-Related Macular Degeneration before Visual Function Is Impaired"

_jcm, 2022, doi:10.3390/jcm11195726_

Round 1

Reviewer 1 Report

This manuscript reveals a cohort of W AMD patients with long-term follow-up. Here are some comments that might add more clarifications. 

  1. Are there any real-world study in the region, this should be stated in introduction and emphasize the gap for why this study was conducted. 
  2. Authors should discuss the VA changes in the long term, significant findings and compare them with the finding from other real-world studies in other region. Does the duration affect VA?
  3. Page 2 line 65: The author states that informed consent was obtained from all patients involved in the study.While, This is a large retrospective case series, does inform consent been requested for all patients? how about the censored patient who dropped out and did not come to follow up at the time of starting the review? 
  4. Please put a reference for converting validation of landolt C chart to LogMar ( if applicable) 
  5. What is the clinician determination regimen, how many ophthalmologists in service? Because this might affect the final result of the treatment.
  6. Patients with PDT / LASER should be adjusted as a factor associated to the final outcome. 
  7. The nonparametric evaluation were used which means that there might be some non-normal distribution of the data. Non-parametric description of the data should be added in the statistical description. 
  8. What is the percentage of successful treatment? 
  9. What is the result of univariable analysis? Short summary description should be fine. 
  10. Is the sample size large enough to adjust multiple variables? Please discuss if there is a sample size calculation and the possible power of detection. 
  11. While logistic regression adjusted for age, cox hazard doesn't use this confounder for adjustment. Please state the reason and justification for confounder selection in both models. 
  12. KM curve , It would be nice if you put a table of N and number of dropouts/censor during time underneath the chart. 
  13. What is the number of injections and clinical activities during the course? Should it be implement for the analysis of the outcome? 
  14. Are there any differences in outcome among the type of anti-VEGF? 
  15. What is the characteristic of a dropout patient? Eg. worse VA patients might give up on the anti-VEGF injection. 

Author Response

Point-by-point responses to the reviewers’ comments

Reviewer 1

This manuscript reveals a cohort of W AMD patients with long-term follow-up. Here are some comments that might add more clarifications.

Thank you for reviewing our manuscript and constructive advice.

  1. Are there any real-world study in the region, this should be stated in introduction and emphasize the gap for why this study was conducted.

Thank you for your comment. We revised the introduction, adding references of previous reports in the region, and emphasized the gap for why this study was conducted. The revised parts are highlighted in yellow as follows;

Line 47

Although, both treatment regimens require judgement of disease activity by individual clinicians, and the determinations of initiation and additional treatment are not unified among clinicians, recent real-world studies show that the treatment is mostly successful[9-13]. However, the analyses had been done using data of patients who had been fully followed-up during the course, and many of those who had dropped-out had been excluded; this may not necessarily represent the real-world practice.

Here we retrospectively reviewed the visual outcomes of eyes with AMD treated with anti-VEGF drugs in a real-world setting. The current study involved all patients who received anti-VEGF treatment as first therapy for AMD and were followed up for at least 12 months and up to 36 months, including those who dropped out during the study period; patients who dropped out were included in Kaplan–Meier survival analyses, which may more precisely reflect real-world conditions of the daily clinical practice.

  1. Authors should discuss the VA changes in the long term, significant findings and compare them with the finding from other real-world studies in other region. Does the duration affect VA?

 Thank you for your comment. Yes, duration may affect VA, as we have shown in Kaplan-Meier analysis; patients whose BCVA decreased > 0.2 in logMAR increased with time. We added discussion as follows;

Line 218

Consistently, significant BCVA improvement was achieved after anti-VEGF treatments in our previous retrospective study of patients with PCV treated by pro re nata regimen at 2 years[9], and another study of patients with neovascular AMD treated by treat-and-extend regimen at 3 years[14]. However, BCVA improvement was not significant in a multicenter retrospective study at 3 years while it was significant at 1 year[15], and less than 1 ETDRS letter improvement was obtained in another multicenter study at 6 years[16]. Because the Kaplan-Meier analysis in the current study showed that the patients whose BCVA decreased > 0.2 in logMAR increased over time, longer follow-up study would be of value in the future.

  1. Page 2 line 65: The author states that informed consent was obtained from all patients involved in the study.While, This is a large retrospective case series, does inform consent been requested for all patients? how about the censored patient who dropped out and did not come to follow up at the time of starting the review? 

The informed consent to use the data for researches and to exclude the data when the patients deny was obtained from all the patients at the first visit in the hospital.

  1. Please put a reference for converting validation of landolt C chart to LogMar ( if applicable)

Thank you for your comment. We converted to the LogMAR score by calculating log(10) of the decimal score obtained by Landolt C chart. We described this point in the revised manuscript as follows;

Line 82

BCVA measured using Landolt C charts was converted into logarithm of the minimum angle of resolution (logMAR) by calculating log(10) of the decimal score obtained by Landolt C chart for the purposes of statistical analyses.

  1. What is the clinician determination regimen, how many ophthalmologists in service? Because this might affect the final result of the treatment.

There were 7 doctors and PRN regimen was mainly used. The limitation regarding this point has already been described in the discussion part in the original manuscript as follows;

Line 304

The limitations of the current study included a relatively small number of patients, retrospective analysis design, involving eyes treated with either or both ranibizumab and aflibercept, and that additional treatment was determined by each clinician. However, this would reflect real-world clinical practice conditions involving variations in clinicians’ determinations over treatment, and the data would be informative to general clinicians.

  1. Patients with PDT / LASER should be adjusted as a factor associated to the final outcome. 

There were 1 patient who underwent PDT, and another patient who underwent micropulse laser during the follow-up as described in the original manuscript (line 94), and influence was minimal. As regards the Kaplan-Meier analysis, these 2 patients had been defined as failure at the time of PDT/laser treatments in the original manuscript.

  1. The nonparametric evaluation were used which means that there might be some non-normal distribution of the dat Non-parametric description of the data should be added in the statistical description.

According to your advice, we added description as follows;

Line 106

Non-parametric analyses were performed.

  1. What is the percentage of successful treatment? 

We added this point in the results as follows;

Line 118

Maintained or improved BCVA at the final visit ≥ 0.2 in logMAR compared with initial BCVA was observed in 93 eyes (89.4%).

  1. What is the result of univariable analysis? Short summary description should be fine.

 Thank you for your comment. The univariate analysis showed that the BCVA < 0.3 in logMAR at baseline was a positive factor (OR, 6.417; 95% CI, 1.345–30.59; P = 0.002), and age (OR, 0.930; 95%CI, 0.871-0.994; P = 0.032) and the presence of fibrovascular PED was a negative factor (OR,0.154; 95% CI, 0.0458–0.528; P = 0.003).

  1. Is the sample size large enough to adjust multiple variables? Please discuss if there is a sample size calculation and the possible power of detection.

There was no sample size calculation. Because the sample size was relatively low, we adjusted only age. We inserted this point in the limitation paragraph as follows;

Line 308

We adjusted only age in the logistic regression analyses considering the sample size.

  1. While logistic regression adjusted for age, cox hazard doesn't use this confounder for adjustment. Please state the reason and justification for confounder selection in both models.

Thank you for your comment. We revised Table 4 and text to show Cox hazard ratio adjusted for age, in the revised manuscript as follows;

Line 22 (abstract)

Kaplan–Meier survival analysis showed that eyes with initial BCVA<0.3 (Cox hazard ratio=2.947; 95%CI=1.047–8.289; P=0.041) had a better survival rate after adjusted for age when failure was defined as a BCVA reduction≥0.2 in logMAR.

Line 181

The baseline characteristics of the eyes related to survival or failure were analyzed using Cox proportional hazard models adjusted for age (Table 4). Initial BCVA < 0.3 in logMAR (hazard ratio, 2.947; 95% CI, 1.047–8.289; P = 0.041) was related to better survival probability.

Hazard ratio

95% Confidence Interval

P

Sex (Male)

0.965

0.389–2.393

0.939

Best-corrected visual acuity < 0.3 in logMAR

2.947

1.047–8.289

0.041*

Polypoidal choroidal vasculopathy (PCV)

1.930

0.747–4.988

0.175

Central retinal thickness ≥ 300 μm

1.325

0.563–3.123

0.519

Intraretinal fluid (IRF)

0.841

0.321–2.204

0.724

Sub-macular hemorrhage (> 1 disc diameter)

0.480

0.161–1.432

0.188

Table 4. Cox hazard ratios of factors associated with post-therapeutic maintained or improved best-corrected visual acuity.

Cox proportional hazard models adjusted for age. *P < 0.05.

  1. KM curve , It would be nice if you put a table of N and number of dropouts/censor during time underneath the chart.

Thank you for your advice. We added the number of the analyzed patients in the Figure 2 and revised the text as follows;

Line 157

The number of eyes analyzed at each time point was shown below the graph of the Kaplan–Meier survival analysis (Figure 2).

Figure 2. Kaplan–Meier survival analysis after anti-vascular endothelial growth factor treatment in age-related macular degeneration. Failure was defined as a best corrected visual acuity (BCVA) reduction ≥ 0.2 in logarithm of the minimum angle of resolution (logMAR) from the initial BCVA. The eyes with initial BCVA better than 0.3 in logMAR (shown in orange) had better survival rate than that of those with initial BCVA equal or worse than 0.3 (shown in blue). P=0.014 by logrank test. The number of the eyes at each time point was shown.

  1. What is the number of injections and clinical activities during the course? Should it be implement for the analysis of the outcome?

Thank you for your comment. We added the number of the injections in the revised manuscript as follows;

Line 119

Mean injection number was 7.5 ±0.5 (range 1 to 22).

As regards implement of injection number in the outcome analysis, respective numbers of the patients during the first 12 months and until the last visit were 5.0 ± 0.2 and 7.6 ± 0.5 in the eyes with maintained or improved BCVA, and 4.9 ± 1.0 (P=0.745) and 6.0 ± 1.0 (P=0.510) in the other eyes, and there were no differences. No other treatments were performed during the course, except for one patients who underwent PDT and another patient who underwent laser, as we described in the response to the comment 6.

  1. Are there any differences in outcome among the type of anti-VEGF?

Thank you for your comment. Because some of the patients used both drugs, we did not analyze this point in the current study and added in the limitation paragraph as follows;

Line 309

Because some of the patients used both drugs, we did not analyze the results according to the drug selection.

  1. What is the characteristic of a dropout patient? Eg. worse VA patients might give up on the anti-VEGF injection.

Thank you for your comment. We checked the BCVA at baseline and at the last visit before drop-out, and found that drop-out patients more frequently had worse BCVA than 0.5 in logMAR at both times. We added a table as revised Table 3, and revised the text as follows;

Line 158

Patients who dopped out by 36 months more frequently involved those whose BCVA > 0.5 at baseline (P=0.039) and BCVA > 0.5 at the last visit (P=0.025) (Table 3). There were no differences in age and sex between the patients who have dropped out or not.

Table 3. Characteristics of the patients who dropped out before month 36.

Dropped out

Continued

P

n (Eyes (%))

23 (22.1)

81 (77.9)

Age (mean ± standard deviation (range))

75.5±2.5

(56-91)

73.4±1.2

(50-93)

0.361

Sex (male %)

14 (60.8)

55 (67.9)

0.866

BCVA at baseline >0.5 (Eyes (%))

13 (56.5)

27 (33.3)

0.039*

BCVA at last visit >0.5 (Eyes (%))

11 (47.8)

18 (22.2)

0.025*

Mann-Whitney U tests and Chi-square tests. BCVA, best-corrected visual acuity. *P < 0.05.

Reviewer 2 Report

In this paper, the authors study impact of treating age-related macular degeneration before visual function is impaired.

As indicated in the introduction, this pathology affects more and more people and current treatments are not equally effective for all patients.

In the material and methods section, the criteria for inclusion and exclusion of patients are not well defined, Could you detail them better?

In the results section, it indicates that the age range of the patients is 50/93 years. Isn't that a too wide range to assess the efficacy of the treatment? Clarify this point, please.

Were all patients treated by the same doctor? How was the most appropriate treatment determined for each patient?

In order to compare the results of the treatment, how did you determine that the baseline characteristics of each patient were the same?

Author Response

Point-by-point responses to the reviewers’ comments

Reviewer 2

In this paper, the authors study impact of treating age-related macular degeneration before visual function is impaired.

As indicated in the introduction, this pathology affects more and more people and current treatments are not equally effective for all patients.

Thank you for reviewing our manuscript and constructive advice.

In the material and methods section, the criteria for inclusion and exclusion of patients are not well defined, Could you detail them better?

Thank you very much for your advice. We revised the manuscript as follows;

Line 71

Medical charts of patients diagnosed with neovascular AMD between January 2017 and August 2020 in the Department of Ophthalmology, St. Luke’s International Hospital (Tokyo, Japan), treated with anti-VEGF drugs as the first therapy for AMD, and followed up for longer than 12 months were retrospectively analyzed. Those who had treated by other therapy than anti-VEGF drugs such as SF6 injection for massive subretinal hemorrhage, or no data of fluorescein angiography for AMD diagnosis were excluded. The data were collected up to 36 months of follow up. There were no patients with both eyes treated during this period.

In the results section, it indicates that the age range of the patients is 50/93 years. Isn't that a too wide range to assess the efficacy of the treatment? Clarify this point, please.

Thank you for your comment. This is a real-world retrospective data based on daily practice, and it is not a too wide range. Clinicians have to treat both relatively younger and older patients in the clinical practice. In addition, Phase III study of faricimab for AMD involved patients who were 55 to 96 years old (Heier et al Lancet 2022), and this is similar to the current study.

Were all patients treated by the same doctor? How was the most appropriate treatment determined for each patient?

There were 7 doctors and PRN regimen was mainly used. The limitation regarding this point has already been described in the discussion part in the original manuscript as follows;

Line 304

The limitations of the current study included a relatively small number of patients, retrospective analysis design, involving eyes treated with either or both ranibizumab and aflibercept, and that additional treatment was determined by each clinician. However, this would reflect real-world clinical practice conditions involving variations in clinicians’ determinations over treatment, and the data would be informative to general clinicians.

In order to compare the results of the treatment, how did you determine that the baseline characteristics of each patient were the same?

We analyzed the results of the treatment and evaluated the differences of the baseline characteristics between survived and failed patients, and between those who had BCVA < 0.3 or ≥ 0.3 at baseline. We did not determine that the baseline characteristics of each patient were the same.

Round 2

Reviewer 1 Report

The authors have addressed all of my comments.